# An Electrochemical Sensor Based on a Porous Biochar/Cuprous Oxide (BC/Cu_2_O) Composite for the Determination of Hg(II)

**DOI:** 10.3390/molecules28145352

**Published:** 2023-07-12

**Authors:** Jin Zou, Jiawei Liu, Guanwei Peng, Haiyan Huang, Linyu Wang, Limin Lu, Yansha Gao, Dongnan Hu, Shangxing Chen

**Affiliations:** 1East China Woody Fragrance and Flavor Engineering Research Center of NF&GA, College of Forestry, Jiangxi Agricultural University, Nanchang 330045, China; 15797710413@163.com; 2Key Laboratory of Chemical Utilization of Plant Resources of Nanchang, College of Chemistry and Materials, Jiangxi Agricultural University, Nanchang 330045, China; 15180018556@163.com (J.L.); pengguanwei163@163.com (G.P.); 13767967571@163.com (H.H.); linyuwang@jxau.edu.cn (L.W.); lulimin816@126.com (L.L.)

**Keywords:** electroanalytical sensor, mercury ion, biochar, cuprous oxide, differential pulse anodic stripping voltammetry

## Abstract

Mercuric ion (Hg^2+^) in aqueous media is extremely toxic to the environment and organisms. Therefore, the ultra-trace electrochemical determination of Hg^2+^ in the environment is of critical importance. In this work, a new electrochemical Hg^2+^ sensing platform based on porous activated carbon (BC/Cu_2_O) modified with cuprous oxide was developed using a simple impregnation pyrolysis method. Differential pulse anodic stripping voltammetry (DPASV) was used to investigate the sensing capability of the BC/Cu_2_O electrode towards Hg^2+^. Due to the excellent conductivity and large specific surface area of BC, and the excellent catalytic activity of Cu_2_O nanoparticles, the prepared BC/Cu_2_O electrode exhibited excellent electrochemical activity. The high sensitivity of the proposed system resulted in a low detection limit of 0.3 ng·L^−1^ and a wide linear response in the ranges from 1.0 ng·L^−1^ to 1.0 mg·L^−1^. In addition, this sensor was found to have good accuracy, acceptable precision, and reproducibility. All of these results show that the BC/Cu_2_O composite is a promising material for Hg^2+^ electrochemical detection.

## 1. Introduction

Heavy metals released through anthropogenic activities, such as industrial, domestic, and technological activities, are of increasing concern [1]. Mercury ion (Hg^2+^), is one of the most hazardous heavy metals that is widely distributed in the environment and can be found in water, air, and soil. Since Hg^2+^ has a high affinity for thiol groups to form Hg-S chemical bonds, it can cause severe damage to the brain, kidneys, the central nervous system, the immune system, and the endocrine system [2,3,4]. Moreover, the accumulation of inorganic Hg^2+^ in the human body can cause toxicity even at low concentrations [5]. According to the report of the World Health Organization (WHO), the maximum level of Hg^2+^ in drinking water is 1 μg·L^–1^ [6]. Therefore, the development of highly sensitive methods for Hg^2+^ determination is of great importance.

For the analysis of traces of mercury, there are accurate and precise methods such as atomic emission spectrometry (AES), mass spectrometry (MS), atomic absorption spectrometry (AAS), and surface-enhanced Raman spectroscopy (SERS) methods [7,8,9,10]. Despite the high accuracy and sensitivity of these methods, the disadvantages of high cost, time-consuming and expensive instrumentation, and the fact that they require professional personnel limit their widespread application. In comparison, electrochemical sensors have recently attracted the attention of numerous researchers for the rapid determination of Hg^2+^ because of their high sensitivity, favorable selectivity, rapid analysis, portability, and inexpensiveness [11,12,13]. In the development of electrochemical sensors, the immobilization of materials with different structures and rich active sites on the electrode surface is advantageous to improve the stability, sensitivity, and repeatability of the sensor.

In recent decades, transition metal oxides have become a hot spot for electrode materials due to their complicated chemical composition, high catalytic activity, and favorable chemical stability [14,15,16]. For example, Baghayeri et al. detected cadmium and lead using an electrochemical probe with glutathione functionalized magnetic nanoparticle (GSH@Fe_3_O_4_) [17]. Pang et al. synthesized Co/CoO/Co_3_O_4_ nanocomposites using a modified hydrothermal microwave carbon bath method, which was used to detect Cu(II), Hg(II), and Cd(II), with detection sensitivities of 5.35 (Cd), 12.71 (Cu), and 4.03 (Hg) μA·μM^−1^, respectively [18]. The metal oxides with unique electrocatalytic interfaces promote the adsorption of HMIs and greatly improve the performance of electroanalysis for the detection of HMIs. However, poor electrical conductivity and easy aggregation limit their usefulness in sensing technology [19,20]. To improve the electron transfer properties, many efforts have been made to incorporate metal oxides into carbon materials. Compared with expensive artificial carbon produced through complex processes, carbon materials produced from biomass as raw material have attracted great attention with their outstanding physical and chemical properties, including widespread sources, low cost, easy production, sustainable development, and highly porous structure [21,22]. In hybrid materials, biochar is considered an important medium for electron transport. In addition, the loading of biochar with metal oxide particles enhances the electrocatalytic activity and prevents the agglomeration of metal oxide particles. Therefore, the unique properties of integrating biochar with metal oxide hybrid materials make them attractive for electrochemical performance.

In this work, a porous composite of biochar/cuprous oxide (BC/Cu_2_O) was synthesized using an impregnation pyrolysis strategy for the sensitive and selective detection of Hg^2+^. The morphological and structural analyses of BC/Cu_2_O were conducted using different characterization methods. The electrochemical characteristics based on different materials were studied via cyclic voltammetry (CV) and electrochemical impedance spectroscopy (EIS). Combining the high catalytic activity and stability of Cu_2_O with the high specific surface specific, porous nanostructure, and charge-transfer ability of BC, the as-synthesized BC/Cu_2_O exhibited high electrocatalytic activity toward Hg^2+^. The current strategy shows excellent sensing performance, including a wide linear range, low detection limit, good selectivity, and high stability. Moreover, the experimental results of Hg^2+^ detection in the lettuce sample demonstrate the feasibility of this method.

## 2. Results and Discussion

### 2.1. Morphology and Structural Characterization

SEM images were used to analyze the morphology of BC and BC/Cu_2_O composite materials. Figure 1B shows the SEM image of BC. The numerous mesopores and macropores indicate that KOH activation resulted in a porous structure that provides numerous active sites for Cu_2_O NP incorporation. The SEM of the BC/Cu_2_O composite is shown in Figure 1C. Compared with pure BC, BC/Cu_2_O still exhibited a typical 3D network structure, and the Cu_2_O NPs were uniformly fixed on the surface of BC. This result confirms the successful synthesis of BC/Cu_2_O.

The crystallographic structures of BC, Cu_2_O, and BC/Cu_2_O were investigated using the XRD diffraction technique (Figure 1D). The XRD phase of BC exhibited important diffraction peaks at 2θ = 25.1° and 43.4°, which correspond to the (002) and (100) diffraction planes of the graphite lattice and are probably related to the formation of the graphite structure [23]. For Cu_2_O, four distinct peaks at 2θ = 37.2°, 43.5°, 50.6°, and 74.3° represent diffraction planes (110), (111), (200), and (220), respectively, which is in good agreement with the JCPDS card of Cu_2_O (JCPDS65-3288) [24]. Interestingly, the diffraction peaks of Cu_2_O and BC were found in BC/Cu_2_O composites. This proves the successful fabrication of BC/Cu_2_O and that the addition of BC did not result in any changes in the crystalline phase of Cu_2_O.

XPS spectroscopy analysis was performed to further investigate the elemental composition and chemical state of the BC/Cu_2_O composite. Figure 2A shows the C 1s, O 1s, and Cu 2p spectra of BC/Cu_2_O. As shown in Figure 2B, the high C 1s spectrum can be divided into four peaks at 284.76, 286.0, 288.16, and 290.34 eV, corresponding to C=C, C−C, C=O, and π–π*, respectively [25]. The XPS spectrum of Cu 2p showed two characteristic peaks at 932.50 eV and 952.36 eV, which can be attributed to Cu 2p_3/2_ and Cu 2p_1/2_ of Cu^+^, respectively (Figure 2C) [26]. Typically, two peaks at 934.38 eV and 954.51 eV are attributed to satellite line peaks. As shown in Figure 2D, the XPS of O 1s centered at 531.71, 533.01, and 534.2 eV can be attributed to C=O, C−O, and adsorbed water molecules, respectively. Moreover, the XPS peak at 530.52 eV is associated with O^2-^ surrounded by Cu atoms [27].

The thermal gravimetric analysis (TGA) of the BC/Cu_2_O composite was performed to estimate the thermal stability of the composites. The temperature range of the weight loss process was from 0 °C to 800 °C in a N_2_ atmosphere. As illustrated in Figure 3A, the TG curve of BC/Cu_2_O demonstrates that BC/Cu_2_O started its weight loss at around 120 °C, possibly caused by the loss of OH/O-terminated groups. The material mass did not show any significant change after 500 °C, which proves the outstanding structural stability of BC/Cu_2_O for applications in a wide temperature range. In addition, the BC/Cu_2_O material underwent an approximately 10.24% weight loss, which is attributed to the Cu_2_O NPs, indicating a high Cu content in the composite, and this is beneficial for electrochemical sensing.

The group structure of Cu_2_O, BC, and BC/Cu_2_O were further analyzed using FT-IR spectra. As shown in Figure 3B, the FT-IR spectrum of Cu_2_O revealed absorption bands at 574 and 683 cm^−1^, which correspond to the Cu−O stretching vibration. The absorption band at 3465 cm^−1^ suggests the presence of −OH functionality due to the moisture present in the as-prepared Cu_2_O nanoparticles. For BC, the absorption band at 3459 cm^−1^ suggests the presence of −OH functional groups. The broad -OH group can be ascribed to the hydroxyl group of the −COOH functional group present in the BC due to the oxidation of surface aliphatic −CH groups in the graphite starting material. The absorption band at 1635 cm^−1^ appeared due to the presence of C=C stretching vibration in the graphitic core. The weaker absorption bands at 614 cm^−1^ and 689 cm^−1^ of the BC/Cu_2_O composite (relative to that of Cu_2_O) suggest that oxygen in BC may be bound to the oxygen-deficient sites of Cu_2_O and that BC are covalently bonded to Cu_2_O via oxygen to form nanohybrid structures.

### 2.2. Electrochemical Characterization of Different Modified Electrodes

The CV plots for BC/Cu_2_O/GCE in a 5.0 mM [Fe(CN)_6_]^3−/4−^ solution with 0.1 M KCl at scan rates ranging from 10 to 80 mV·s^−1^ are shown in Figure 4. The electrochemical surface area can be derived according to the Randles–Sevick equation [28]:I_p_ = 2.69 × 10^5^ *n*^3/2^ A_eff_ D_0_^1/2^ C *v*^1/2^(1)
where I_p_ is the peak current value. A represents the effective area of the working electrode. C represents the [Fe(CN)_6_]^3−/4−^ concentration. *v* represents the scan rate. D (7.6 × 10^−6^ cm^2^·s^−1^) and *n* (*n* = 1) represent the diffusion coefficient and the number of electrons involved in the redox reaction, respectively. Using Equation (1), it was found that the active surface area of the BC/Cu_2_O/GCE was 0.0803 cm^2^, which is higher than that of GCE (0.0707 cm^2^), indicating the superior electrocatalytic activity of BC/Cu_2_O composite in the redox reaction of Hg^2+^.

The inherent interfacial properties of bare GCE, BC/GCE, and BC/Cu_2_O/GCE were measured using EIS studies. Figure 5A shows the Nyquist plot of bare GCE, BC/GCE, and BC/Cu_2_O/GCE. As can be seen, there are clear semicircle curves in the EIS spectra of bare GCE, BC/GCE, and BC/Cu_2_O/GCE. The inset in Figure 4A shows the Randle equivalent circuit, which is composed of the Warburg constant (Zw), the resistance of the solution (Rs), electron-transfer resistance (Ret), and the double-layer capacitance (Cdl). For bare GCE, the Ret value was found to be 652.3 Ω. After modification with BC, the observed low Ret (58.13 Ω) value indicates the high electrical conductivity of BC. The modification of GCE with BC/Cu_2_O increased the electron-transfer resistance compared with BC/GCE, which is due to the low electrical conductivity of Cu_2_O.

The electrochemical behavior of 100 μg·L^−1^ Hg^2+^ on bare GCE, BC/GCE, and BC/Cu_2_O/GCE was investigated using DPASV in 0.1 M of ABS solution. As shown in Figure 5B, bare GCE presented an obvious striping peak of Hg^2+^ at 0.2 V. Compared with bare GCE, the DPASV signal of BC/GCE slightly increased, which may be attributed to outstanding electrical conductivity and large specific surface area of BC. Furthermore, the GCE modified with BC/Cu_2_O showed the highest peak current. The unique sensing performance mainly originates from the fact that Cu_2_O possesses a synergistic adsorption effect with BC to enhance the response current of Hg^2+^.

### 2.3. Optimization of Analytical Parameters

To obtain the best response for the detection of Hg^2+^, the optimization of pH, deposition potential, deposition time, and material volume is very important.

pH optimization was performed in the range of pH 4.0–6.0 in 0.1 mol·L^−1^ of ABS buffer (Figure 6A). The DPASV peak current of Hg^2+^ gradually increased from pH 4.0 to 5.0. At a low pH, the solution contained more hydrogen ions, which can compete with the heavy metal ions for the active site on the electrode surface. When the pH increased further, the anodic peak current decreased sharply due to the hydrolysis of the metal ions [29]. The peak current of Hg^2+^ dissolution reached its maximum at pH = 5.0. So, a pH of 5.0 was used for the next stripping determinations.

The effect of deposition time (150–270 s) on the 100 μg·L^−1^ Hg^2+^ detection efficiency is shown in Figure 6B. As shown, the stripping peak current rapidly increased from 120 to 210 s, which is due to the increased amount of Hg^2+^ on the modified electrode surface. Then, the stripping peak current flattened from 210 to 270 s, which is attributed to the saturated deposition of Hg^2+^ on the BC/Cu_2_O/GCE surface. Considering the balance between the detection sensitivity and the measurement efficiency, the optimum deposition time was 210 s.

The effect of deposition potential on Hg^2+^ signals was investigated in the potential range from −0.6 V to −1.0 V (Figure 6C). The peak current of Hg^2+^ increased significantly at first and then gradually decreased with a negative shift of the deposition potential. When the deposition potential is too negative, the hydrogen evolution reaction could destroy the metal ions deposited on the electrode surface, resulting in a decrease in the dissolution peak current [30]. In addition, the more positive the deposition potential is, the more difficult it is for the reduction reaction of Hg^2+^ to occur, reducing the accumulation of Hg^0^ on the electrode surface. The maximum response current of Hg^2+^ was observed at a deposition potential of −0.8 V. Therefore, −0.8 V was established as the optimum deposition potential for Hg^2+^ detection.

The effect of BC/Cu_2_O dispersion volume on Hg^2+^ current was evaluated as 1–9 μL. As shown in Figure 6D, the peak current response of Hg^2+^ increased significantly with the suspension volume from 1 to 5 μL, which is due to the fact that there were more active sites for Hg^2+^ adsorption with increasing BC/Cu_2_O loading. However, the peak current decreased significantly after 5 μL, which is due to the large amount of BC/Cu_2_O that hindered electron transfer at the electrode surface. With the material volume of 5 μL, the current intensity reached its maximum. Therefore, the volume of BC/Cu_2_O dispersion was set at 5 μL in subsequent studies.

### 2.4. DPASV Determination of Hg^2+^

Under the optimal experimental conditions, the electrocatalytic activity of the BC/Cu_2_O-based sensor toward Hg^2+^ was evaluated using DPASV. In Figure 7A, a clear peak belonging to Hg^2+^ can be observed at about 0.18 V. Meanwhile, the current response of Hg^2+^ increased with increasing Hg^2+^ concentration. As shown in Figure 7B, the I_p_ value is linear with the Hg^2+^ concentration in the range of 1.0 ng·L^−1^ to 1.0 mg·L^−1.^ The corresponding linear relationship is I_p_ (μA) = 0.0179 + 0.05 C (μg·L^−1^) (R_2_ = 0.9975), and the LOD is calculated as 0.3 ng·L^−1^ (S/N = 3). From Table 1, it can be shown that the proposed sensor displays better analytical performance than that reported in the literature [31,32,33,34]. This remarkable property might be due to the excellent electron transfer capability, the large specific surface area, and the excellent catalytic activity of the BC/Cu_2_O composite. Therefore, the proposed BC/Cu_2_O composite is expected to be a promising candidate for the detection of Hg^2+^.

### 2.5. Repeatability, Reproducibility, Stability, and Selectivity Experiments

Under optimal experimental conditions, the repeatability, reproducibility, and storage stability of BC/Cu_2_O/GCE were evaluated using the DPASV technique. The repeatability of BC/Cu_2_O/GCE was examined by detecting 100 μg·L^−1^ Hg^2+^ for 15 tests continuously (Figure 8A). The oxidation current did not display larger transformations and the relative standard deviation (RSD) was 1.53%, indicating that BC/Cu_2_O/GCE had a satisfying repeatability. In addition, the RSD of parallel detection of 100 μg·L^−1^ Hg^2+^ with six parallel BC/Cu_2_O/GCEs was 3.62%, indicating the exceptional reproducibility of the preparation of BC/Cu_2_O/GCE.

The storage stability of BC/Cu_2_O/GCE was studied with the addition of 100 μg·L^−1^ Hg^2+^ for 15 days. As shown in Figure 8C, the DPASV response for Hg^2+^ retained 92.83% of the current on that first day, suggesting the excellent stability of BC/Cu_2_O/GCE. As a result, the BC/Cu_2_O/GCE sensor was found to have superior repeatability, reproducibility, and stability.

To assess the selective sensing ability of BC/Cu_2_O/GCE, interference tests were performed by measuring the DPASV of 100 μg·L^−1^ Hg^2+^ with various interference matrices. To this end, 100-fold excess concentrations of Pb^2+^, Cd^2+^, Cu^2+^, Mg^2+^, Na^+^, Zn^2+^, Mn^2+^ were used as interference studies. Under optimal experimental conditions, DPASV was used to determine the electrochemical response of Hg^2+^ alone and in the mixture. As shown in Figure 8D, the presence of interferents had a negligible impact on the detection of Hg^2+^ (signal variation was less than 5%), indicating that the BC/Cu_2_O/GCE sensor has good anti-interference ability for the determination of Hg^2+^.

### 2.6. Performance Evaluation for Practical Application

The practicality of BC/Cu_2_O/GCE was evaluated for the determination of Hg^2+^ in a lettuce sample. Lettuce was purchased from a local vegetable market, which was chopped and then homogenized in a juicer for 5 min. Afterward, the acquired lettuce juice was further centrifuged and diluted with 0.1 M ABS (pH = 5.0). Under optimal experimental conditions, Hg^2+^ standard solutions were spiked into the lettuce sample and detected using DPASV. As shown in Table 2, the recoveries were calculated in the range of 99.8 to 104.3% with RSD values of 3.81–4.43%. The obtained recovery results indicate that the prepared BC/Cu_2_O/GCE can be used as an accurate and efficient sensing platform for Hg^2+^ detection in real samples.

## 3. Experimental Section

### 3.1. Reagents and Chemicals

Potassium hydroxide (KOH, 99.5%), zirconium tetrachloride (ZrCl_4_, 98%), ethanol (CH_3_CH_2_OH, 98%), copper nitrate (Cu(NO_3_)_2_·5H_2_O, ≥99%), and mercury nitrate (Hg(NO_3_)_2_, ≥99%) were purchased from Sigma-Aldrich. An acetate buffer solution (ABS, 0.1 M, pH 5.0) was prepared by mixing 0.1 M of sodium acetate and acetic acid. All chemicals are of analytical grade and used without further purification. All the reagent solutions and buffer solutions were prepared using ultra-pure water (>18 MX, Millipore, Burlington, MA, USA).

### 3.2. Instruments

All electrochemical experiments were carried out on a CHI 660E electrochemical workstation (Shanghai Chenhua, Shanghai, China), including electrochemical impedance spectroscopy (EIS) and differential pulse stripping voltammetry (DPASV), with a three-electrode system. A saturated calomel electrode (SCE) and a platinum wire electrode were used as reference electrodes and counter electrodes, respectively. Bare or material-decorated glass carbon electrode (GCE, its diameter is 3 mm) served as the working electrode. X-ray diffraction (XRD, Rigaku D/MAX-2500 powder, Beijing, China) measurements of products were characterized with an X-ray diffractometer using Cu Kα radiation (λ = 1.5418 Å) at 40 kV and 40 mA in the 2θ region of 10–50°. The morphology and dimensions of the samples were obtained via scanning electron microscopy (SEM, Hitachi S4800, Tokyo, Japan). The electronic states of the surfaces of the samples were analyzed using X-ray photoelectron spectroscopy (XPS, Escalab 250, Shanghai, China). Thermogravimetric analysis (TGA) was performed using a Q50 Universal V20.13 Build 39 (TA instruments, New Castle, DE, USA). The sample was tested using a ramp profile (10 °C/min) from 30 °C to 800 °C under N_2_. The functional groups in the materials were analyzed with a Fourier transform infrared (FT-IR) spectrophotometer procured from Bruker, Hamburg, Germany. The sample was mixed with KBr and crushed into pellets before the test. The scan range was fixed at 400–4000 cm^−1^. 

### 3.3. Preparation of Cu_2_O/BC Composite

BC derived from Litsea cubeba was produced by slowly pyrolyzing the feedstock under Ar protection. First, the waste Litsea cubeba samples were washed with deionized (DI) water repeatedly to remove physical contaminants on the surface. Subsequently, 2.8 g of Litsea cubeba was dispersed in a KOH solution (1.0 mol·L^−1^, 100 mL). After that, the obtained suspension was sealed in a 150 mL hydrothermal synthesis reactor, followed by a hydrothermal process (150 °C, 6 h). After drying at 80 °C overnight, the activated Litsea cubeba was carbonized at 800 °C for 2 h under an Ar atmosphere, with a heating rate of 5 °C·min^−1^. When cooled down to room temperature, the activated BC was ultrasonically washed with 1 mol·L^−1^ of HCl and deionized water in sequence until pH = 7. After drying and grinding, BC was obtained.

Briefly, 1 g of BC was immersed in 35 mL of a KOH solution (1 M) containing 1 g of Cu(NO_3_)_2_.5H_2_O. After being sonicated for 10 min, the mixture was soaked at room temperature for 24 h. Then, the BC rich in Cu^2+^ ions was dried in a 60 °C drying oven and transferred into a tube furnace. The mixture was heated at 500 °C for 2 h (temperature rise 5 °C/min) in an Ar flow. Finally, the obtained material was washed with 1 mol·L^−1^ HCl and deionized water until neutral.

### 3.4. Preparation of Working Electrode

First, 3 mg of BC/Cu_2_O was dispersed into 3 mL of deionized water to prepare a 1 mg·mL^−1^ BC/Cu_2_O suspension with the aid of ultrasonication. Before electrochemical testing, bare GCE was carefully polished using 0.05 μm of alumina aqueous slurry and then successively rinsed with ethanol and distilled water. After drying at room temperature, 5 μL of the BC/Cu_2_O hybrid material solution was spread onto the surface of the GCE and dried at room temperature. The prepared electrode was denoted as BC/Cu_2_O/GCE for further electrochemical research. As a comparison, BC/GCE and Cu_2_O/GCE were also prepared using the same procedure, respectively.

### 3.5. Measurement of Electrochemical Properties

In order to carry out electrochemical sensing of Hg^2+^, the modified electrode was immersed into 0.1 M of an acetate buffer solution (pH 5.0) containing Hg^2+^ with varying concentrations under optimized conditions. DPASV was applied at a deposition potential of −0.8 V for 210 s and was performed in the potential range of 0.10 to 0.26 V under the following conditions: an increment potential of 4 mV, a frequency of 15 Hz, an amplitude of 50 mV, and a pulse width of 50 ms. Prior to the next cycle, BC/Cu_2_O/GCE was dipped in a fresh supporting electrolyte at a potential of +0.6 V for 180 s in order to remove the residual metal ions on the surface of the working electrode. All measurements were performed at room temperature. The preparation procedure for the BC/Cu_2_O/GCE composite and the DPASV method for Hg^2+^ detection is shown in Figure 1A.

Electrochemical impedance spectroscopy experiments were performed in a solution including 5 mM of [Fe(CN)_6_]^3−/4−^ and 0.1 M of KCl at the formal potential of 0.25 V, with a frequency changing from 0.01 Hz to 100 kHz and a signal amplitude of 5 mV.

## 4. Conclusions

In this work, an electrochemical sensor of BC/Cu_2_O composites was successfully developed for Hg^2+^ detection. Benefitting from the synergistic effects of enrichment capacity, strong catalytic activity, high conductivity, and large specific surface area, the BC/Cu_2_O composite material exhibited excellent electrochemical performance. Using the optimal analytical parameters, the BC/Cu_2_O/GCE sensor possesses a wider linear range, high reproducibility, and good practical application performance. In addition, the proposed sensor was validated for Hg^2+^ detection in a real sample with very satisfactory outcomes. Thus, the BC/Cu_2_O/GCE has great application potential as a sensitive and reliable electrochemical platform for the determination of Hg^2+^ in food samples.

## Figures and Tables

**Figure 1 molecules-28-05352-f001:**
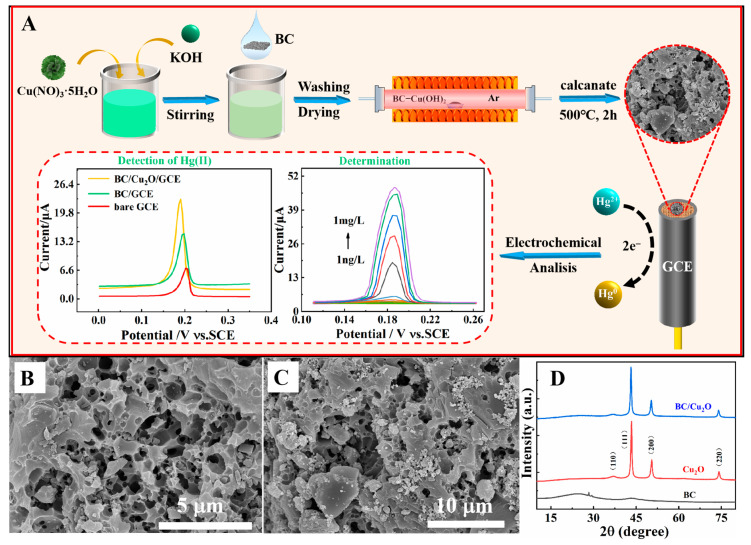
(**A**) Synthesis route of BC/Cu_2_O and its applications in Hg^2+^ detection; SEM images of (**B**) BC, (**C**) BC/Cu_2_O; and (**D**) XRD of BC, Cu_2_O, and BC/Cu_2_O composite.

**Figure 2 molecules-28-05352-f002:**
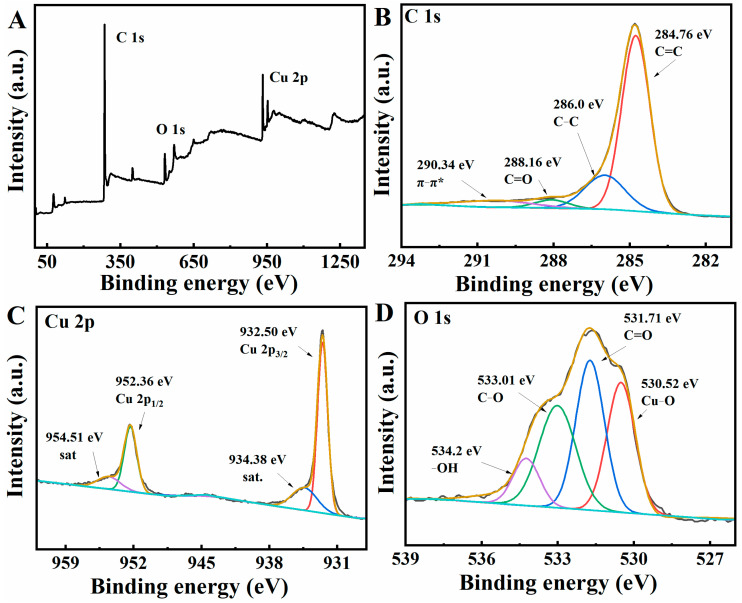
XPS diffraction patterns for BC/Cu_2_O: full scan (**A**), high-resolution C 1s spectrum (**B**), high-resolution Cu 2p spectrum (**C**), and high-resolution O 1s spectrum (**D**).

**Figure 3 molecules-28-05352-f003:**
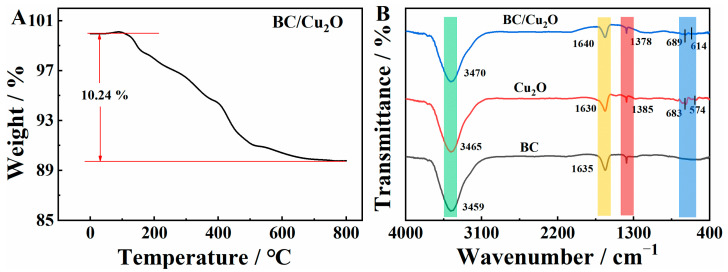
(**A**) TGA curve of the BC/Cu_2_O composite; (**B**) FT-IR spectra of BC, Cu_2_O, and BC/Cu_2_O.

**Figure 4 molecules-28-05352-f004:**
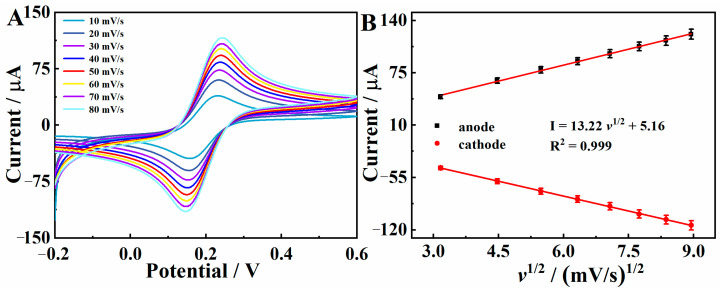
In 5 mM [Fe(CN)_6_]^3−/4−^ and 0.1 M KCl mixture solution, (**A**) CV curves of the BC/Cu_2_O at different scan rates ranging from 10 to 80 mV/s; (**B**) the corresponding linear relationship between the redox peak currents and the square root of scan rates for BC/Cu_2_O.

**Figure 5 molecules-28-05352-f005:**
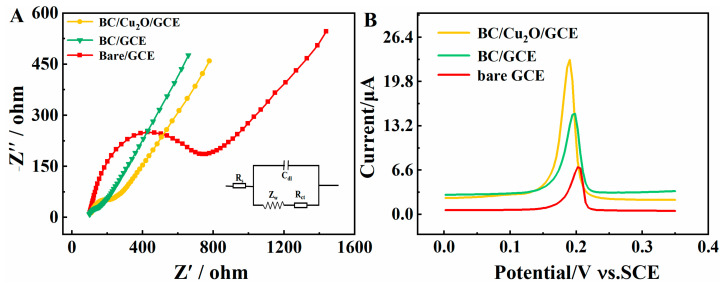
(**A**) Electrochemical impedance spectroscopy of bare GCE, BC/GCE, and BC/Cu_2_O/GCE; (**B**) DPASV of 100 μg·L^−1^ Hg^2+^ at bare GCE, BC/GCE, and BC/Cu_2_O/GCE in 0.1 mol·L^−1^ acetate buffer solution (pH = 5.0).

**Figure 6 molecules-28-05352-f006:**
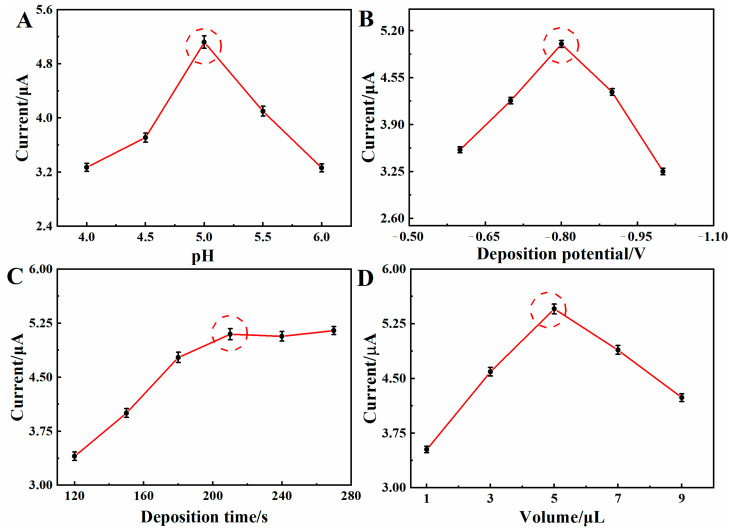
Optimization of experimental conditions. Influence of (**A**) pH value of the detection solution, (**B**) deposition potential, (**C**) deposition time, and (**D**) material volume on the stripping peak current of Hg^2+^ on the BC/Cu_2_O/GCE.

**Figure 7 molecules-28-05352-f007:**
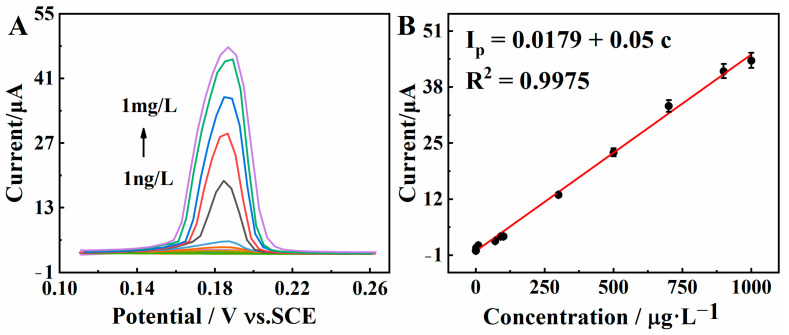
(**A**) DPASV stripping signals of the BC/Cu_2_O/GCE for the detection of Hg^2+^ over a concentration range from 1.0 ng·L^−1^ to 1.0 mg·L^−1^; (**B**) the linear plots of the stripping peak current density versus concentrations of Hg^2+^.

**Figure 8 molecules-28-05352-f008:**
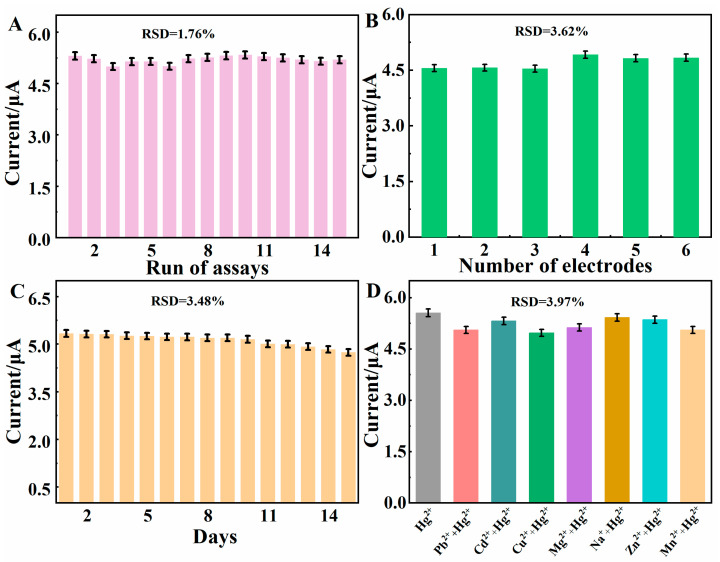
(**A**) Repeatability measurements for one BC/Cu_2_O/GCE; (**B**) reproducibility measurements at six independent BC/Cu_2_O/GCEs; (**C**) stability measurements at BC/Cu_2_O/GCE within 15 days; (**D**) selectivity of BC/Cu_2_O/GCE.

**Table 1 molecules-28-05352-t001:** Comparison of the analytical performance of various reported electrochemical sensors for measurement of Hg^2+^.

Electrode Substrate	Measurement Technique	Linear Range (μg·L^−1^)	LOD (μg·L^−1^)	References
rGOS ^a^@SnO_2_	DPV	50.15–141,476	16.72	[31]
Fe_3_O_4_/F-MWCNTs ^b^	SWV	2.61–6519	0.78	[32]
ZnO/rGO/Ppy ^c^	DPV	0.40–7.22	0.13	[33]
GCE/Cu-MOF	DPV	0.02–10.03	0.013	[34]
ZrO_2_/N-3DPC ^d^	DPASV	0.1–220	0.062	[35]
BC/Cu_2_O	DPASV	0.001–1000	0.0003	This work

Note: ^a^: reduced graphene oxide; ^b^: fluorinated multiwalled carbon nanotubes; ^c^: polypyrrole; ^d^: nitrogen-doped three-dimensional porous carbon.

**Table 2 molecules-28-05352-t002:** Recoveries of trace Hg^2+^ in real lettuce sample (*n* = 3).

Sample	Added (μg·L^−1^)	Founded (μg·L^−1^)	RSD (%)	Recovery (%)
1	0	0	0	0
2	0.1	0.107	3.81	102.7
3	0.3	0.301	4.43	104.3
4	0.5	0.449	3.39	99.8

## Data Availability

The data presented in this study are available in the article.

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
