# Peer review of "An Electrochemical Sensor Based on a Porous Biochar/Cuprous Oxide (BC/Cu2O) Composite for the Determination of Hg(II)"

_molecules, 2023, doi:10.3390/molecules28145352_

Round 1

Reviewer 1 Report

Reviewer’s suggestions and comments on the Manuscript entitled:

Electrochemical sensor based on porous biochar/cuprous oxide (BC/Cu2O) composite for the determination of Hg (II)

Manuscript ID: molecules-2479278

Authors have successfully developed an electrochemical sensor of BC/Cu2O composites for Hg2+ detection. It has been shown that the BC/Cu2O composite material exhibits excellent electrochemical performance based on the synergistic effects. Results obtained suggest high reproducibility and good practical application performance on a wide linear range. The investigated sensor was validated for Hg2+ detection in the real sample. Overall, the BC/Cu2O/GCE sensor has a bright application potential for determining Hg2+ in real samples.

This manuscript has citation potency. Therefore I strongly recommend that the Editorial Office accept this manuscript after minor revision.

Suggestions:

-          Line 95 Missing ° (sign for degrees)

-          Line 122 Equation has to be numbered left from the equation.

-          Line 126 Is it equation 2? Where is Eq. 1?

Author Response

Point 1: Line 95 Missing ° (sign for degrees).

Response 1: Thanks to reviewers for their kind suggestions. The symbol of “°” have been added in our revised manuscript [p.3, line 138].

Point 2: Line 122 Equation has to be numbered left from the equation.

Response 2: Thanks to reviewers for their kind suggestions. Equation has been numbered left from the equation in our revised manuscript [p.6, line 208].

Point 3: Line 126, is it equation 2? Where is Eq. 1?

Response 3: We are grateful to the reviewer for pointing out our mistake. Equation has been numbered left from the equation and only equation 1 exist in our revised manuscript [p.6, line 208].

Again, thanks a lot for the kind help and comments from the editor and the reviewers. Wish you a pleasant and fruitful future.

Reviewer 2 Report

The paper describes the detection of Hg2+ using porous biochar/cuprous oxide 2 (BC/Cu2O) composite. There is a novelty in this work. Authors have used porous biochar/cuprous oxide 2 (BC/Cu2O) composite. The paper provides relevant data to the study of detection of Hg2+ , however at this stage the text has misleading information, and has several errors (tipos) that have to be corrected. Some of the interpretations made need further support data or have to be corrected. The manuscript seems to be suitable for publishing if revised (minor revision) according to the comments presented below.

1-      Why the authors used Differential pulse anodic stripping voltammetry (DPASV)?

2-      Why the authors used porous biochar/cuprous oxide 2 (BC/Cu2O) composite?

3-      The authors need to talk about thermal stability of porous biochar/cuprous oxide 2 (BC/Cu2O) composite.

4-      The author needs to explain why choose Hg2+

5-      Should be a single space between the number and unite.

6-      In the literature review, the author should be mention  the following work:

·         Voltammetric Determination of Hg2+, Zn2+, and Pb2+ Ions Using a PEDOT/NTA-Modified Electrode..  https://pubs.acs.org/doi/full/10.1021/acsomega.2c02682, https://doi.org/10.1021/acsomega.2c02682

·         Use of a Schiff base-modified conducting polymer electrode for electrochemical assay of Cd (II) and Pb (II) ions by square wave voltammetry.. https://link.springer.com/article/10.1007/s11696-021-01882-7 , https://doi.org/10.1007/s11696-021-01882-7

7-      Are the Pt electrode home made or commercial?

8-       In  conclusion, the author should be clearly mentioned what are the best conditions for detection  and why.

9-       There are many grammar and spilling mistakes.

There are some grammar and spilling mistakes.

Author Response

Point 1: Why the authors used Differential pulse anodic stripping voltammetry (DPASV)?

Response 1: Thanks to reviewers for their kind suggestions. Differential pulse anodic stripping voltammetry (DPASV) is a powerful tool for the determination of trace heavy metal ions via the combination of a credible preconcentration step with accurate electrochemical measurement of the accumulated analytes.

Point 2: Why the authors used porous biochar/cuprous oxide (BC/Cu2O) composite?

Response 2: Thanks to reviewers for their kind suggestions. In recent decades, transition metal oxides have become popular electrode materials owing to their high catalytic activity, low-cost, complicated chemical composition, and favorable chemical stability. Among them, cuprous oxide (Cu2O) has received significant attention due to its large surface to volume ratio, high catalytic activity and non-toxicity. In addition, the unique electrocatalytic interfaces of Cu2O will promote the adsorption of HMIs and greatly improve the detection performance for HMIs. However, the poor electrical conductivity and easy aggregation limit their application in electrochemical sensing.

Porous carbon (BC) materials derived from biomass has attracted extensive attention due to their outstanding physical and chemical properties, including widespread source, low cost, easy preparation, sustainable development, large specific surface area, high porosity and excellent electrical conductivity. Here, combining Cu2O with BC will not only effectively avoid their agglomeration, making the adsorption and catalytic sites fully exposed, but also enhance their electron transport capacity. The synergistic effect between Cu2O and BC endowed the BC/Cu2O composite with superior detection properties for HMIs.

Hence, in the present work, BC/Cu2O composite was synthesized through an impregnation pyrolysis strategy for the sensitive and selective detection of Hg2+.

Point 3: The authors need to talk about thermal stability of porous biochar/cuprous oxide (BC/Cu2O) composite.

Response 3: Thanks for the reviewer’s kind suggestion. The thermal gravimetric analysis of BC/Cu2O has been added in our revised manuscript [p. 5, line 169-181].

The thermal gravimetric analysis (TGA) of BC/Cu2O composite was performed to estimate the thermal stability of the composites. The temperature range of the weight loss process is from 0 °C to 800 °C in a N2 atmosphere. As illustrated in Fig. 3A, the TG curve of BC/Cu2O demonstrates that BC/Cu2O started its weight loss at around 120 ℃, possibly caused by the loss of OH/O terminated groups. The material mass did not show any significant change after 500 ℃, which proves the outstanding structural stability of BC/Cu2O for applications in a wide temperature range. In addition, the BC/Cu2O material shows about an 10.24% weight loss, ascribing to the Cu2O NPs, indicate a high Cu content in the composite, and this is beneficial for electrochemical sensing.

Fig. 3. (A) TGA curve of the BC/Cu2O composite; (B) FT-IR spectra of BC, Cu2O, and BC/Cu2O.

Point 4: The author needs to explain why choose Hg2+?

Response 4: Thanks for the reviewer’s kind suggestion. In fact, the BC/Cu2O electrode has electrochemical response toward four metal ions of Cu2+, Hg2+, Pb2+ and Cd2+. As shown in the following figure, four distinct stripping peaks for Cu2+ (100 μg/L), Hg2+ (100 μg/L), Pb2+ (100 μg/L) and Cd2+ (100 μg/L) were observed at -0.06 V, 0.18 V, -0.57 V and -0.80 V, respectively (Fig. R1A). However, by comparing with Fig. R1B, the peaks of Cu2+ and Cu2O are found to overlap. In addition, compared with Pb2+ and Cd2+, the detection of Hg2+ is more sensitive. Considering the above reasons, in the work, Hg2+ is chosen as detection target. Since the stripping peaks for Cu2+, Hg2+, Pb2+ and Cd2+ can be well distinguished, Hg2+ can be selectively detected at BC/Cu2O electrode.

Fig. R1. (A) DPASV stripping signals for simultaneous detection of 0.09 μg/mL Cd2+, Pb2+, Cu2+ and Hg2+ at BC/Cu2O/GCE; (B) blank detection in the absence of heavy metal ions at BC/Cu2O/GCE.

Point 5: Should be a single space between the number and unite.

Response 5: Thanks to reviewers for their kind suggestions. Single space between the number and unite have been added in our whole revised manuscript.

Point 6: In the literature review, the author should be mentioning the following work: Voltammetric Determination of Hg2+, Zn2+, and Pb2+ Ions Using a PEDOT/NTA-Modified Electrode. https://pubs.acs.org/doi/full/10.1021/acsomega. 2c02682, https://doi.org/10.1021/ acsomega.2c02682; Use of a Schiff base-modified conducting polymer electrode for electrochemical assay of Cd (II) and Pb (II) ions by square wave voltammetry. https://link.springer.com/article/ 10.1007/s11696-021-01882-7, https://doi.org/10.1007/s11696-021-01882-7. 

Response 6: Thanks to reviewers for their kind suggestions. These references have been cited in our revised manuscript [p.17, line 600-605].

Point 7: Is the Pt electrode home made or commercial?

Response 7: Thanks to reviewers for their kind suggestions. The Pt electrode is commercial.

Point 8: In conclusion, the author should be clearly mentioned what are the best conditions for detection and why. 

Response 8: Thanks to reviewers for their kind suggestions. The supporting electrolyte pH, deposition potential, deposition time and material volume for Hg2+ detection are 5.0, -0.8 V, 210 s, and 5 μL, respectively. The best conditions for detection and reasons have been complemented in our revised manuscript [p.7, line 254-299].

Point 9: There are many grammar and spilling mistakes. 

Response 9: Thanks to reviewers for their kind suggestions. The grammar and spilling mistakes have been corrected in our whole revised manuscript.

Again, thanks a lot for the kind help and comments from the editor and the reviewers. Wish you a pleasant and fruitful future.

Reviewer 3 Report

This manuscript by Zou et al. described a biosensor for detecting Hg2+ by using a composite electrode. The sensor shown good detection limit and selectivity, as well as a wide linear range. The authors also verify the function of this sensor in real lettuce sample. This manuscript is generally well-written and the results sounds. Recommend to publish after addressing the following issues.

1. By using spray coating, how the authors make sure the adhesion of BC/CU2O on GCE electrode is good?

2. In line 127, the authors should point our the surface area of bare GCE electrode.

3. In line 145, the voltage is 0.2V instead of 0.18V?

4. Why BC/CU2O/GCE shows higher peak than BC/GCE in Figure 4B considering it has higher charge transfer resistance than BC/GCE as indicated in Figure 4A?

5. Figure labelling after section 2.3 are all wrong, please correct.

6. Can the author explain the mechanism of selectivity?

The whole manuscript is generally well-written.

Author Response

Point 1: By using spray coating, how the authors make sure the adhesion of BC/Cu2O on GCE electrode is good?

Response 1: Thanks to reviewers for their kind suggestions. In spray coating, good adhesion between the BC/Cu2O and the GCE can be achieved by following two aspects: on the one hand, the glassy carbon electrode surface should be carefully cleaned and polished prior to spray coating to ensure that it is free of any contaminants, such as oils and dust particles, that may hinder adhesion. On the other hand, functional groups, such as -OH and -COOH, on BC/Cu2O surface can form hydrogen bonds or covalent bonds with GCE surface, which enhances adhesion.

Point 2: In line 127, the authors should point out the surface area of bare GCE electrode.

Response 2: Thanks to reviewers for their kind suggestions. The surface area of bare GCE electrode is 0.0707 cm2. This information has been added in our revised manuscript [p.6, line 216].

Point 3: In line 145, the voltage is 0.2 V instead of 0.18 V?

Response 3: Thanks to reviewers for their kind suggestions. The voltage has been corrected to 0.2 V in our revised manuscript [p.7, line 240].

Point 4: Why BC/Cu2O/GCE shows higher peak than BC/GCE in Figure 4B considering it has higher charge transfer resistance than BC/GCE as indicated in Figure 4A?

Response 4: Thanks to reviewers for their kind suggestions. Usually, electrochemical impedance spectroscopy serves as a mean to verify the conductivity of modified electrodes. From Figure 4A, it can be observed that BC/GCE exhibits superior conductivity compared to BC/Cu2O/GCE, which is consistent with the background current of Hg2+ detection in Figure 4B for BC/GCE and BC/Cu2O/GCE. However, the sensing performance of modified electrodes for Hg2+ detection not only relies on the material conductivity, but also greatly depends on the effective surface area and catalytic ability of the materials, which play important roles in enhancing the detection performance of the modified electrode. In this work, the poor conductivity of Cu2O reduces the charge transfer ability of the BC/Cu2O composite, while its large specific surface area, high catalytic activity, and unique electrocatalytic interface promotes the adsorption and catalytic of Hg2+ and greatly improves the detection performance of Hg2+.

Point 5: Figure labelling after section 2.3 is all wrong, please correct.

Response 5: Thanks to reviewers for their kind suggestions. Figure labelling after section 2.3 have been all corrected in our revised manuscript.

Point 6: Can the author explain the mechanism of selectivity? 

Response 6: Thanks for the reviewer’s kind suggestion. To evaluate the selectivity of BC/Cu2O/GCE on the detection of Hg2+, 100-fold excess concentration of Pb2+, Cd2+, Cu2+, Mg2+, Na+, Zn2+, Mn2+ were added into the cell containing 100 μg·L-1 Hg2+. As can be seen from the following figure, after the injection of different interferent ions, slight changes in peak current is observed for Hg2+ (signal variation less than 5%). This phenomenon might be attributed to the moderate competition for active adsorption sites at the BC/Cu2O/GCE by the interfering ions [1,2].

[1] S. Lee, J. Oh, D. Kim, Y.Z. Piao, A sensitive electrochemical sensor using an iron oxide/graphene composite for the simultaneous detection of heavy metal ions. Talanta. 160 (2016) 528-536. http://dx.doi.org/10.1016/j.talanta.2016.07.034.

[2] N. Ruecha, N. Rodthongkum, D.M. Cate, J. Volckens, O. Chailapakul, C.S. Henry, Sensitive electrochemical sensor using a graphene-polyaniline nanocomposite for simultaneous detection of Zn(II), Cd(II), and Pb(II). Anal. Chim. Acta. 874 (2015) 40-48. http://dx.doi.org/10.1016/j.aca.2015.02.064.

Again, thanks a lot for the kind help and comments from the editor and the reviewers. Wish you a pleasant and fruitful future.